# H2 URESONIC: Design of a Solar-Hydrogen University Renewable Energy System for a New and Innovative Campus

**Salaki Reynaldo Joshua ***, **Sanguk Park *** and **Kihyeon Kwon ***

Department of Electronics, Information and Communication Engineering, Kangwon National University, Samcheok-si 25913, Republic of Korea
* Correspondence: joshua@kangwon.ac.kr (S.R.J.); sanguk@kangwon.ac.kr (S.P.); kweon@kangwon.ac.kr (K.K.)

**Abstract:** The necessity to move to sustainable energy solutions has inspired an investigation of innovative technologies for satisfying educational institutions' sustainable energy needs. The possibility of a solar-hydrogen storage system and its integration into university energy management is investigated in this article. The study opens by providing context, noting the growing relevance of renewable energy in universities as well as the necessity for effective energy storage systems. The goal is to delve into solar-hydrogen technology, outlining its components, operating mechanism, and benefits over typical storage systems. The chapter on Integration Design examines current university energy infrastructure, identifies problems, and provides ways for integrating solar-hydrogen systems seamlessly. This integration relies heavily on technological and economic considerations, such as a cost-benefit analysis and scalability studies. Case studies include real-world examples, performance measurements, and significant insights learned from successful implementations. The chapter Future Prospects investigates new trends in solar-hydrogen technology as well as the impact of government legislation, providing a forward-looking viewpoint for colleges considering adoption. The report concludes with a summary of significant findings, emphasizing the benefits of solar-hydrogen integration and making recommendations for future implementations. The limitation of this research is that it only focuses on design and simulation as a phase of preliminary study.

**Keywords:** solar-hydrogen storage system; university energy management; renewable energy; energy storage technologies; sustainable energy

## 1. Introduction

The global shift toward sustainable and renewable energy sources has acquired tremendous traction in recent years [1]. Universities, as centers of research and learning, increasingly understand the importance of embracing environmentally sustainable energy solutions [2]. This paradigm shift is motivated not only by environmental concerns but also by the possibility for educational institutions to serve as models for sustainable practices [3]. In today's energy consumption situation, the need to shift toward more sustainable methods is becoming increasingly clear [4]. Traditional reliance on nonrenewable energy sources has exacerbated environmental challenges, demanding a rethinking of energy management practices [5,6]. In this perspective, the integration of solar-hydrogen storage systems appears to be a promising answer to universities' energy concerns. Several strong reasons motivate us to conduct research into the integration of solar-hydrogen storage devices into university energy management (Table 1) [7–9].

In essence, research into the integration of solar-hydrogen storage systems is consistent with a broader commitment to environmental stewardship, fiscal restraint, and the instructional purpose of institutions [10]. It positions institutions as innovators in the adoption of revolutionary technologies that not only meet their energy requirements but also contribute to a more sustainable and resilient future [11].

**Table 1.** The Implementation of Solar-Hydrogen Energy in Universities.

| No | Indicator | Description |
|---|---|---|
| 1 | Environmental Imperative | The worldwide imperative to combat climate change necessitates a transition toward more environmentally friendly energy practices. Universities, as significant institutions, must set a good example in terms of decreasing carbon footprints and supporting ecologically friendly technologies. |
| 2 | Economic Viability | Renewable energy technologies, such as solar-hydrogen systems, provide enormous economic benefits. Exploring and adopting these solutions can result in long-term cost reductions, making it a financially sound decision for institutions. |
| 3 | Educational Leadership | Universities act as educational leaders and social influencers. They not only contribute to environmental sustainability by embracing innovative energy solutions, but they also teach and motivate the next generation of leaders to embrace eco-friendly activities. |
| 4 | Energy Resilience | Diversifying energy sources by implementing solar-hydrogen storage systems improves energy resilience. Universities can benefit from a secure and sustainable energy supply because of their vital responsibilities in research, education, and community service. |
| 5 | Technological Advancements | As technology advances, remaining on the cutting edge of energy storage solutions is critical. Solar-hydrogen system research allows universities to take advantage of cutting-edge technology, ensuring that their energy infrastructure remains efficient and competitive. |
| 6 | Regulatory and Policy Landscape | Renewable energy initiatives are increasingly favored by government laws and regulations. Researching and implementing solar-hydrogen storage systems allows institutions to take advantage of favorable legislation, which may result in financial incentives and awards. |
| 7 | Community and Stakeholder Expectations | Universities are vital components of their communities. Adopting sustainable energy methods corresponds to the expectations of environmentally concerned stakeholders, such as students, teachers, staff, and the broader community. |
| 8 | Long-Term Sustainability | Sustainable energy solutions help institutions' long-term viability and sustainability. Institutions can protect their energy infrastructure from potential resource shortages or price volatility by lowering their reliance on traditional energy sources. |

The development of solar-hydrogen energy presents a viable and sustainable approach to address the negative environmental effects associated with traditional energy sources. This method encourages the production of carbon-neutral energy by using solar electricity to produce hydrogen via electrolysis of water. When operating, solar-hydrogen systems emit no greenhouse gases, in contrast to fossil fuels, which helps to reduce air pollution and mitigate climate change. Additionally, by providing a flexible and environmentally benign substitute in industries like transportation, manufacturing, and power generation, hydrogen as a clean energy carrier aids in the shift to a low-carbon economy. Adopting solar-hydrogen energy supports a cleaner and more sustainable energy future and is in line with international efforts to stop environmental damage [12].

Solar-hydrogen energy combines solar power with electrolysis to produce hydrogen, thereby emulating a clean and sustainable energy paradigm. Sunlight is captured by solar

panels and transformed into electricity, which is then used to split water molecules to produce hydrogen, an environmentally friendly fuel. This technology offers a carbon-neutral substitute for traditional energy sources because it produces no emissions at all. The intermittent nature of solar electricity can be mitigated with hydrogen, an energy carrier that is both flexible and storable. Furthermore, hydrogen only produces water vapor when it is burned or used in fuel cells, confirming its status as a clean and sustainable energy source. This novel combination of solar and hydrogen technologies not only lessens their negative effects on the environment but also establishes solar-hydrogen energy as a key participant in the shift to a more sustainable and environmentally friendly energy landscape [13].

Numerous advantages arise from using electrolytic systems in universities to produce hydrogen, supporting both sustainability and academic goals. Universities can achieve environmental stewardship and lower their carbon footprint by producing clean hydrogen on-site through the use of electrolyzers driven by renewable energy sources like solar or wind. Furthermore, incorporating electrolytic systems into academic environments provides a useful and instructive example of cutting-edge sustainable technology. In order to promote an innovative and environmentally conscious culture, staff and students get personal experience with hydrogen production methods and applications of renewable energy. Moreover, the hydrogen generated may be used as a flexible energy source for a range of campus uses, such as hydrogen fuel for automobiles, offering a real-world illustration of how sustainable energy solutions can be incorporated into regular operations [14].

The current study examines the scientific and financial feasibility of powering Delhi Technological University's Science Block in Delhi, India, utilizing solar photovoltaic energy sources. In a hybrid energy system, the intermittent solar energy is stabilized by using hydrogen energy storage to produce a steady electrical current. The most ideal design, with a net current cost of $1,030,406, has been simulated by HOMER software. With a rated capacity of 240 kW and a mean output of 44.5 kW—or 1068 kWh/day—solar photovoltaic generates 389.865 MWh of energy every year. When operating for 4344 h a year, the PV system can produce up to 253 kW, with a PV penetration of 358% [15].

The major goal of this research is to offer a thorough examination of the solar-hydrogen storage system. Universities can make informed decisions about adopting and integrating this technology into their energy infrastructure if they grasp its complexities. The research intends to go into the practical issues of integrating solar-hydrogen systems into university energy management, in addition to the theoretical ones. This entails investigating universities' existing energy infrastructure, identifying potential obstacles, and suggesting effective integration options.

Because colleges shape future leaders and decision-makers, their dedication to sustainable energy practices can have a significant impact on societal norms and practices. We hope to contribute to the discussion of sustainable energy in educational institutions with this study, which focuses on the architecture and integration design of solar-hydrogen storage systems.

## 2. Solar-Hydrogen Storage System

### 2.1. Solar-Hydrogen Technologies and Storage System

The solar-hydrogen storage system is an innovative technique for capturing and storing renewable energy [16]. This section presents a comprehensive review of the technology, focusing on the synergy between solar power and hydrogen as an energy carrier. This technology solves the intermittent nature of solar power and promotes efficient energy storage by turning solar energy into electricity and then using electrolysis to make hydrogen [17].

This section examines the unique advantages afforded by solar-hydrogen technology, highlighting the limits of typical energy storage technologies. These benefits include increased storage capacity, scalability, and the ability to store energy for long periods of time. Furthermore, the system's ability to generate a clean energy carrier (hydrogen) makes it an appealing approach to lowering carbon emissions.

Solar-hydrogen technologies cover a wide range of approaches for harnessing solar energy and converting it to hydrogen for storage. Typically, numerous critical components are involved in the integration of various technologies (Table 2) [18–20].

**Table 2.** Key Components of Solar-Hydrogen Technologies.

| No | Technology | Description |
|----|-----------|-------------|
| 1 | Photovoltaic (PV) solar panels | Photovoltaic solar panels use semiconductor elements to convert sunlight directly into electricity. These panels capture solar energy for subsequent conversion and are an essential component of solar-hydrogen systems. When exposed to sunlight, PV panels generate direct current (DC) electricity. |
| 2 | Electrolyzers | Electrolyzers are machines that use electricity to split water ($H_2O$) into hydrogen ($H_2$) and oxygen ($O_2$). There are three types of electrolyzers: Proton Exchange Membrane (PEM) Electrolyzers, which are efficient and well-suited for smaller-scale applications; Alkaline Electrolyzers, which are economical and well-suited for large-scale industrial applications; and Solid Oxide Electrolyzers, which operate at high temperatures and are well-suited for specific industrial processes. |
| 3 | Hydrogen Storage Tanks | Hydrogen storage tanks are used to keep the hydrogen produced for later use. The first type of hydrogen storage is gaseous hydrogen storage, which is stored under pressure in high-strength tanks. The second type of storage is liquid hydrogen storage, which uses extremely low temperatures to convert hydrogen to a liquid state. The third type of storage is solid-state hydrogen storage, which uses materials that absorb and release hydrogen as needed. |
| 4 | Fuel Cells | Fuel cells are electrochemical devices that produce energy and water by converting hydrogen and oxygen. The most prevalent kind of fuel cells are Proton Exchange Membrane (PEM) Fuel Cells, which are commonly utilized for stationary and portable applications. Less prevalent are Alkaline Fuel Cells (AFC), which have historically been utilized in space applications, and Solid Oxide Fuel Cells (SOFC), which operate at high temperatures and are suited for a variety of applications. |
| 5 | Solar Thermal Hydrogen Production | Concentrated solar power is used to generate high-temperature heat for the thermal breakdown of water into hydrogen and oxygen in solar thermal hydrogen synthesis. Solar concentrators focus sunlight onto a receiver, producing high temperatures that fuel chemical processes. |
| 6 | Photoelectrochemical (PEC) Cells | PEC cells use semiconductor materials to directly convert solar energy into hydrogen via a photoelectrochemical process. The PEC function of light absorption in the semiconductor material generates electron-hole pairs, which initiates the water-splitting reaction. |
| 7 | Bifacial Solar Panels | Solar panels that gather sunlight from both the front and back faces capture more energy overall. Increased efficiency by harnessing reflected sunlight from surrounding surfaces is the function. |

*2.2. Components*

2.2.1. Solar Panel

This section looks at how solar panels capture sunlight and transform it into electrical energy. It discusses advances in solar panel technology and its effectiveness in producing

clean electricity. The photovoltaic effect describes the method by which solar panels catch sunlight and convert it into electrical energy (Table 3) [21–23].

**Table 3.** The Process of Solar Panels Capturing Sunlight and Transforming it into Electrical Energy.

| No | Step | Description |
|----|------|-------------|
| 1 | Photons Absorption | When sunlight, which consists of photons (light particles), strikes the surface of a solar panel, the photons transfer their energy to the electrons in the panel's semiconductor material. |
| 2 | Generation of Electron-Hole Pairs | Electrons in the semiconductor material are excited by the absorbed energy and break out from their regular locations. This produces electron-hole pairs, in which one electron is freed and a positively charged "hole" in the substance is left behind. |
| 3 | Creation of Voltage Potential | Electron mobility generates an electric current, and charge separation generates a voltage potential across the solar cell. This potential difference serves as the foundation for generating electricity. |
| 4 | Electricity Generation | The electric current created travels through the semiconductor material and into an external circuit connected to the solar panel. This movement of electrons creates electrical energy, which can then be used for a variety of purposes. |
| 5 | Direct Current (DC) Output | Solar panels generate electrical energy in the form of direct current (DC). DC power flows in only one direction and is created by solar cells. |
| 6 | Inverter Conversion (for Grid-Connected Systems) | An inverter converts the direct current (DC) electricity generated by solar panels into alternating current (AC) for grid-connected solar power systems. AC is the most common type of electricity found in homes and businesses. |
| 7 | Utilization or Grid Feed-in | The converted electrical energy can be used immediately to power devices connected to the solar panel system. Excess electricity can also be fed into the electrical grid, earning credits or compensation in some grid-tied systems. |

The photovoltaic effect is a key phenomenon that enables solar panels to capture sunlight and transform it into a clean, renewable source of electricity [24,25]. Photovoltaic cells made of semiconductor materials like silicon capture incoming photons from sunlight in this complicated operation. This absorption energizes electrons, causing them to be liberated from their typical locations and the formation of electron-hole pairs. The flow of these liberated electrons creates an electric current, which results in a voltage potential across the solar cell [26]. This resulting electrical energy, primarily in the form of direct current (DC), is immediately collected and used to power devices linked to the solar panel system. In grid-connected systems, an inverter may convert direct current (DC) to alternating current (AC) for more versatile uses [27]. Notably, the photovoltaic effect highlights the sustainable nature of solar energy by emitting no emissions and employing a plentiful and renewable resource—sunlight—for a cleaner and more ecologically friendly power generation process [28].

### 2.2.2. Electrolyzer

An electrolyzer is a critical component in the process of producing hydrogen via electrolysis, which uses electrical energy to divide water molecules into hydrogen and oxygen [29]. These devices, which include Proton Exchange Membrane (PEM), alkaline, and solid oxide electrolyzers, are critical in the production of clean and renewable hydrogen [30]. During operation, an electrolyzer receives an electric current, which is often derived from renewable energy sources such as solar or wind power, and applies it to water ($H_2O$). Within the electrolyzer, the electrical energy causes a chemical process that causes water

molecules to separate. During the process, one electrode (cathode) releases hydrogen gas, while the other electrode (anode) releases oxygen gas. The split hydrogen can be stored for subsequent use as a versatile and environmentally beneficial energy carrier, thereby advancing green energy technology and lowering carbon emissions [31].

Factors influencing electrolyzer efficiency and performance include the type of electrolyzer technology used, the cleanliness of the water source, and the overall system design [32]. Technological advancements in electrolyzers continue to improve efficiency, lower prices, and broaden the possibility of large-scale hydrogen generation, establishing electrolysis as a vital actor in the transition to a sustainable and hydrogen-based energy economy [33].

Several factors influence electrolyzer performance and efficiency (Table 4), affecting their capacity to effectively create hydrogen through water electrolysis. Here are some important variables to consider [34–36]:

**Table 4.** Factors influencing electrolyzer performance and efficiency.

| No | Variable | Description |
|----|----------|-------------|
| 1 | Electrolyzer Technology | The efficiency, prices, and applications of various electrolyzer technologies, such as Proton Exchange Membrane (PEM), alkaline, and solid oxide electrolyzers, differ. The technology chosen is determined by considerations such as the anticipated size of hydrogen generation and the application's specific requirements |
| 2 | Electrolyzer Efficiency | An electrolyzer's overall efficiency is critical in calculating the amount of electrical energy required to create a particular amount of hydrogen. Higher efficiency corresponds to lower energy losses throughout the electrolysis process, making hydrogen production more sustainable and cost-effective. |
| 3 | Electrolyzer Dimensions and Capacity | The scalability and usability of the electrolyzer system are influenced by its size and capacity. It is critical to match the size of the electrolyzer to the expected hydrogen output in order to optimize performance and resource utilization. |
| 4 | Purity of Water Source | The quality and purity of the water used in electrolysis affect the performance and longevity of the electrolyzer. Water impurities, such as minerals or pollutants, can degrade electrode performance over time. |
| 5 | Operating Conditions | The efficiency of electrolyzers is affected by operating variables such as temperature, pressure, and flow rates. The optimal working conditions differ based on the type of electrolyzer technology used. |
| 6 | Electrode Materials | The selection of electrode materials has a considerable impact on the electrolyzer's performance and durability. Materials with high conductivity, corrosion resistance, and stability under electrolysis conditions contribute to system longevity and consistency. |
| 7 | Power Supply | The source and stability of the electrical power provided to the electrolyzer have an impact on its performance. Renewable energy sources, such as solar or wind power, contribute to the environmental sustainability of hydrogen generation, while a constant power supply assures consistent operation. |
| 8 | System Design and Integration | The whole design and integration of the electrolyzer system, including the layout of components and control systems, play a role in obtaining optimal performance. Well-designed systems improve efficiency, ease of maintenance, and the overall economic viability of hydrogen production. |

**Table 4.** *Cont.*

| No | Variable | Description |
|---|---|---|
| 9 | Cost Considerations | The cost of electrolyzer systems, including both initial capital expenditures and continuous running expenses, has an impact on the economic feasibility of hydrogen production. Continuous technological breakthroughs and economies of scale are lowering the costs of electrolyzer systems. |
| 10 | Regulatory Environment | The regulatory landscape for hydrogen generation, including regulations, incentives, and standards, can influence the acceptance and deployment of electrolyzer technology. Electrolyzers can be integrated into sustainable hydrogen production systems more quickly if supportive policies and incentives are in place |

As electrolyzer technology advances, addressing these contributing elements becomes critical for increasing efficiency, lowering costs, and fulfilling the potential of hydrogen as a clean and adaptable energy carrier [37].

### 2.2.3. Electrolyzer Hydrogen Storage Tanks

Efficient hydrogen storage is critical for using hydrogen as a versatile energy carrier, allowing it to be integrated into a variety of applications ranging from fuel cells to industrial processes. Several methods and technologies have been developed to securely and efficiently store hydrogen [38–40]:

1. Compressed Hydrogen Storage: gaseous hydrogen is compressed at high pressures, typically ranging from 350 to 700 bar, in compressed hydrogen storage to reduce volume and increase storage density. The compressed hydrogen is kept in high-strength containers made of materials that can resist the pressures involved. While this approach is well-established and very easy, the energy required for compression and the need for durable storage materials are factors in its efficiency. Advanced composite materials and metal hydrides are being investigated to improve the performance of compressed hydrogen storage systems.

2. Liquid Hydrogen Storage: liquid hydrogen storage entails chilling gaseous hydrogen to extremely low temperatures (about −253 degrees Celsius) in order to convert it to a liquid state. This considerably reduces the volume occupied by hydrogen, allowing for increased storage density. Cryogenic storage tanks, sometimes double-walled with vacuum insulation, are used to hold liquid hydrogen. The high energy density achieved by this technology is efficient, but the energy-intensive liquefaction process and the difficulty involved with maintaining low temperatures are aspects to consider.

3. Metal Hydride Storage: metal hydride storage involves the absorption and desorption of hydrogen by specific metal alloys. During absorption, hydrogen is chemically bonded to the metal, and during desorption, hydrogen is liberated. This approach has a distinct benefit in that it provides for safe and compact storage at moderate pressures and temperatures. The effectiveness of metal hydride storage is governed by parameters such as metal alloy selection, operating circumstances, and hydrogen absorption and release kinetics. Ongoing research focuses on building high-capacity, reversible metal hydrides for increased performance.

4. Underground Storage: underground storage, which is commonly found in depleted natural gas reservoirs or salt caverns, makes use of natural geology to store hydrogen. Gaseous hydrogen is injected and stored in subsurface formations, taking advantage of the geological structure's permeability and porosity. This approach offers large-scale storage capacity and is especially well-suited for seasonal storage requirements. Underground storage efficiency is determined by elements such as geological conditions, injection and extraction rates, and safety concerns.

Each storage method has advantages and disadvantages, and the decision is influenced by factors such as the intended application, storage scale, and safety concerns [41]. Ongoing research and development aims to improve the efficiency, safety, and economic viability of hydrogen storage systems, thereby contributing to the wider adoption of hydrogen as a sustainable energy solution [42].

### 2.2.4. Fuel Cells

Fuel cells serve an important role in completing the energy cycle while providing significant efficiency and environmental benefits. These electrochemical systems transform hydrogen's chemical energy directly into electricity, with only water vapor and heat as byproducts. Fuel cell efficiency is a significant advantage over traditional combustion-based energy conversion technologies [43]. Traditional power production systems, such as internal combustion engines, sometimes entail many energy conversion processes, each of which results in energy losses. Fuel cells, on the other hand, are more efficient, turning more than 50% of the energy content in hydrogen into electricity. This simpler energy conversion process improves overall system efficiency by minimizing energy waste and boosting the energy cycle's sustainability [44].

Furthermore, the environmental benefits of fuel cells derive from their clean and emission-free operation. Unlike combustion-based power generation, which emits pollutants and greenhouse gasses, fuel cells generate energy via an electrochemical reaction that does not involve combustion [45]. The principal byproduct of fuel cell operation is water vapor, making them an environmentally favorable option for power generation. This quality is especially important in the context of mitigating climate change and lowering air pollution [46]. As the demand for clean energy solutions grows, fuel cells emerge as a potential technology to address both efficiency difficulties and the environmental consequences associated with traditional energy cycles. Fuel cells' adaptability, from small-scale applications like portable electronics to larger-scale deployments in cars and stationary power systems, positions them as a crucial element in developing a sustainable and low-emission energy future [47].

### 2.3. Working Mechanism

The importance of this painstakingly developed operating mechanism resides in its transformational potential for the energy environment. Converting solar energy to electricity is the first step toward harnessing an abundant and renewable resource, providing a clean and sustainable alternative to traditional power generation [48]. The ensuing electrolysis process for hydrogen generation not only takes advantage of solar-generated electricity, but also creates a green channel for hydrogen—a versatile and efficient energy carrier [49]. The critical role of hydrogen storage and retrieval addresses the intermittent nature of solar energy by allowing excess energy to be stored for later use. Finally, fuel cell energy generation completes the cycle by transforming stored hydrogen into electricity with unparalleled efficiency and with the least possible environmental impact [50]. This entire working mechanism not only propels us toward a more sustainable and resilient energy paradigm, but it also highlights the possibility for integrated, closed-loop systems that align with the principles of environmental stewardship and green energy advancement [51–53]:

- Solar Energy Conversion to Electricity: the process begins with the conversion of solar energy into electricity via photovoltaic (PV) cells. These cells, which are typically composed of semiconductor materials such as silicon, absorb photons from the sun. The energy from the absorbed photons stimulates electrons in the semiconductor, resulting in an electric current. This direct conversion of sunlight into electricity, known as the photovoltaic effect, is the first step in harvesting solar energy for later stages in the energy cycle.
- Electrolysis Process for Hydrogen Production: the power generated by solar panels is then routed to an electrolyzer for electrolysis, which creates hydrogen from water ($H_2O$). Electrical energy is used in the electrolyzer to break water molecules into

hydrogen and oxygen. The electrolysis is facilitated by two electrodes, one connected to the positive terminal (anode) and the other to the negative terminal (cathode). The liberated hydrogen gas is collected, and as a byproduct, oxygen is emitted. The use of solar-generated electricity for electrolysis assures a sustainable and environmentally benign technique of producing hydrogen.

- Hydrogen Storage and Retrieval: once created, hydrogen is efficiently stored for subsequent use. Compression, liquefaction, metal hydrides, and underground storage are all means of storage. The storage mechanism chosen is determined by considerations like scale, application, and accessibility. These hydrogen stores serve as a versatile energy transporter, ready to be retrieved when needed. The ability to store hydrogen allows for greater flexibility in energy distribution and consumption, addressing the intermittent nature of solar energy production.

- Fuel Cell Electricity Generation: the final stage of the energy cycle involves the utilization of hydrogen in fuel cells to generate electricity. Fuel cells work by using an electrochemical reaction between hydrogen and oxygen. In a typical Proton Exchange Membrane (PEM) fuel cell, hydrogen is delivered to the anode, and oxygen from the air is fed to the cathode. Hydrogen molecules are divided into protons and electrons at the anode. Protons pass through a membrane, while electrons generate an electric current. Protons, electrons, and oxygen mix at the cathode to form water and heat as byproducts. This electrochemical technique directly turns the chemical energy of stored hydrogen into electrical energy, completing the energy cycle with high efficiency and low environmental impact. Thus, fuel cells play an important role in delivering a clean and sustainable source of electricity for a range of applications.

The coordinated working mechanisms of solar energy conversion, electrolysis for hydrogen synthesis, hydrogen storage, and fuel cell power generation provide a convincing model for a sustainable and closed-loop energy system. Beginning with the direct conversion of solar energy into electricity via photovoltaic cells, the cycle effortlessly advances to the electrolysis stage, where solar-generated electricity supports the environmentally friendly generation of hydrogen from water [54]. The subsequent storage of hydrogen, using various and effective methods, overcomes the temporal variability inherent in solar energy harvesting. Finally, fuel cell electricity generation completes the cycle by efficiently transforming the chemical energy of stored hydrogen into electricity with low environmental impact. This integrated process not only uses renewable resources to meet energy needs, but it also shows a harmonious interplay of technology, providing a viable path toward a resilient, low-emission, and sustainable energy future [55].

## 3. Research Approach and Integration Design

### 3.1. Research Approach

The research strategy for this study is extensive and interdisciplinary, including theoretical frameworks, empirical studies, and practical applications. The methodology is designed to evaluate the integration of a solar-hydrogen storage system into a university's energy management system, taking into account a variety of factors ranging from technological feasibility to environmental impact.

- Literature Review: the investigation begins with a thorough assessment of the available literature on solar-hydrogen systems, university energy management, and associated technologies. This phase lays the theoretical groundwork by synthesizing essential concepts, best practices, and field difficulties. It also identifies knowledge gaps, which informs research questions and objectives.

- Case Studies and Interviews: analyzing real-world case studies where solar-hydrogen systems have been integrated into university energy management is a critical component. In addition, interviews with experts, stakeholders, and professionals in the industry provide useful insights into practical issues, accomplishments, and lessons gained from previous projects. These qualitative data help to provide a more sophisticated view of the integration process.

- Technology Assessment: the study includes a full evaluation of the technological aspects, including the efficiency, scalability, and economic viability, of solar-hydrogen storage systems. This includes a thorough analysis of several types of solar panels, electrolyzers, storage systems, and fuel cell technologies to decide which are most suited for academic applications.
- Simulation and Modeling: the study evaluates the potential energy generation, storage capacity, and overall system performance under varied situations using simulation tools and modeling approaches. This step tries to provide predicted insights into the system's behavior by taking into account elements such as solar irradiation, swings in energy consumption, and the university's specific energy needs.
- Environmental Impact Analysis: an environmental impact assessment is carried out to determine the integrated system's ecological footprint. Analyzing life cycle evaluations, carbon emissions reductions, and overall sustainability measures is part of this process. It looks at the broader environmental effects of switching to a solar-hydrogen energy system at a university.
- Integration Design and Feasibility Analysis: the study synthesizes findings into a holistic integration design, defining the step-by-step procedure for integrating the solar-hydrogen storage system into the university's existing energy infrastructure. A feasibility analysis looks at the economics, potential obstacles, and overall feasibility of adopting the suggested design.

This interdisciplinary study strategy ensures a comprehensive understanding of the integration process, taking into account technical, economic, and environmental factors. The integration of theoretical concepts, empirical facts, and practical considerations results in a solid and well-informed examination of solar-hydrogen integration within the particular setting of university energy management.

### 3.2. Integration Design

#### 3.2.1. Existing Energy Infrastructure

Energy sustainability takes center stage at this forward-thinking university, which harnesses the power of four separate sources spread over its campus. A diverse energy portfolio includes solar panels, wind turbines, combined heat and power systems, and a grid link. Sources of Energy at Kangwon National University Samcheok Campus (Figure 1):

- Joint Laboratory and Practice Building (Building 123);
- Engineering Building V (Building 120);
- Engineering Building IV (Building 118);
- Engineering Building II (Building 122).

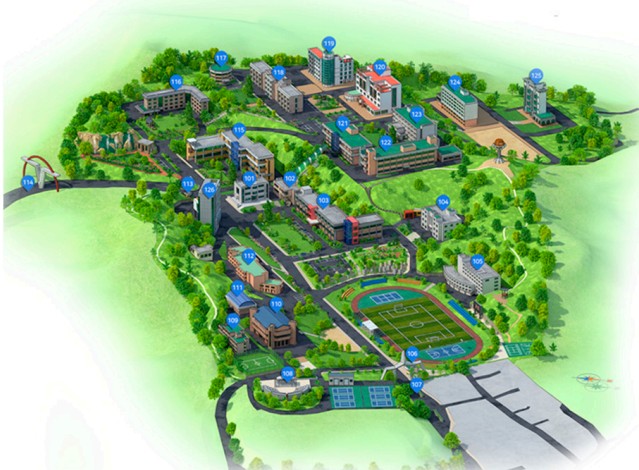

**Figure 1.** Sources of Energy from Kangwon National University Samcheok Campus.

### 3.2.2. Solar Generation Monitoring System

The university receives unique real-time insights into its energy dynamics. This system is distinguished by its integration with a cutting-edge online monitoring platform (Figure 2). This not only allows for more effective management, but also gives stakeholders the option to remotely monitor and alter energy consumption, responding quickly to demand variations or any unforeseen concerns. The web-based monitoring system serves as a transparent lighthouse, enabling administrators and facility managers to make informed decisions, enhance energy efficiency, and demonstrate a commitment to sustainable practices.

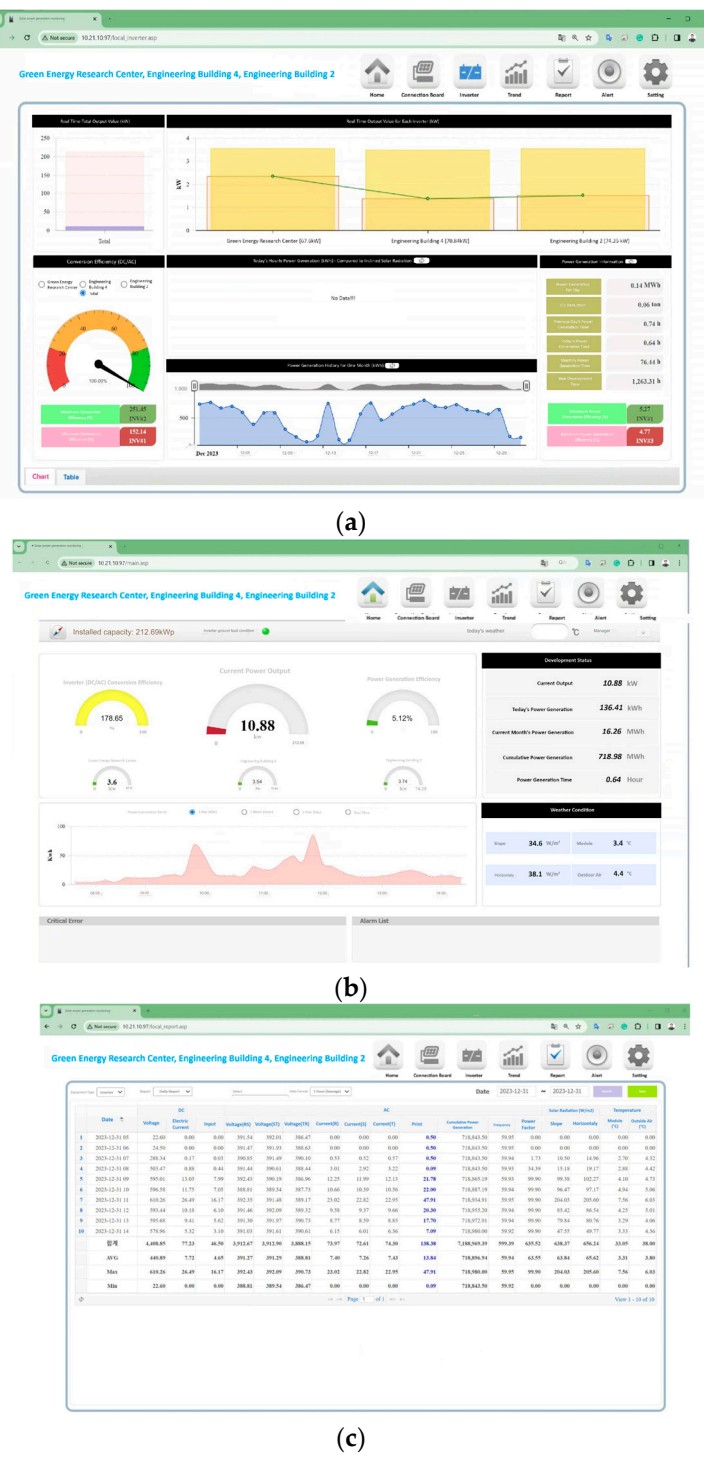

(**a**)

(**b**)

(**c**)

**Figure 2.** *Cont.*

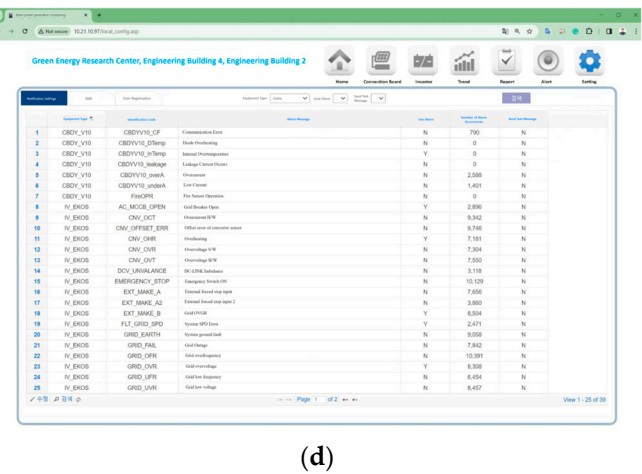

(**d**)

**Figure 2.** Elements of Solar Generation Monitoring System: (**a**) Inverter; (**b**) Home Dashboard; (**c**) Report; (**d**) Settings.

This holistic approach to energy management is consistent with the university's commitment to environmental responsibility and technical innovation. By seamlessly integrating renewable energy sources and providing a user-friendly interface for monitoring and control, the institution not only decreases its carbon footprint but also establishes a precedent for using innovative technology in pursuit of a greener and more efficient campus. The synergy between various energy sources and the web monitoring system presents a picture of a university at the vanguard of sustainable energy practices, producing a campus climate where cutting-edge technology meets environmental care.

## 4. Modeling a Solar-Hydrogen System in the University

### 4.1. Energy Management System Model

The goal of creating the Microgrid EMS (Energy Management System) Optimization for Solar-Hydrogen (Figure 3) comprises a broad set of goals targeted at tackling current energy concerns and advancing sustainable solutions. The major goal is to incorporate solar and hydrogen technologies into a microgrid context. This integration (Figure 4) intends to maximize energy capture and storage by leveraging the complementary nature of these renewable sources. By maximizing energy storage, the concept attempts to improve the dependability and resilience of microgrid systems. The microgrid becomes more resilient in the face of external disruptions or variations by efficiently controlling the storage and delivery of solar and hydrogen-derived electricity.

Designing a Hybrid Solar-Hydrogen model in Simulink that depicts the interactions between an electrolyzer, energy storage components, and solar photovoltaic (PV) system is the first step in designing a hybrid solar-hydrogen system. The PV system, including the panels, inverters, and controllers, can be represented using Simulink's graphical user interface. The electrolyzer stack and hydrogen storage can also be dynamically modeled. The integration of control systems and power electronics enhances system performance, and time-domain simulations evaluate the system's capabilities under different scenarios. Performance metrics and sensitivity analyses are used to assess overall system behavior, efficiency, and hydrogen generation rates. Results can be analyzed with the help of Simulink's visualization tools, and the best operating conditions can be found by applying optimization techniques.

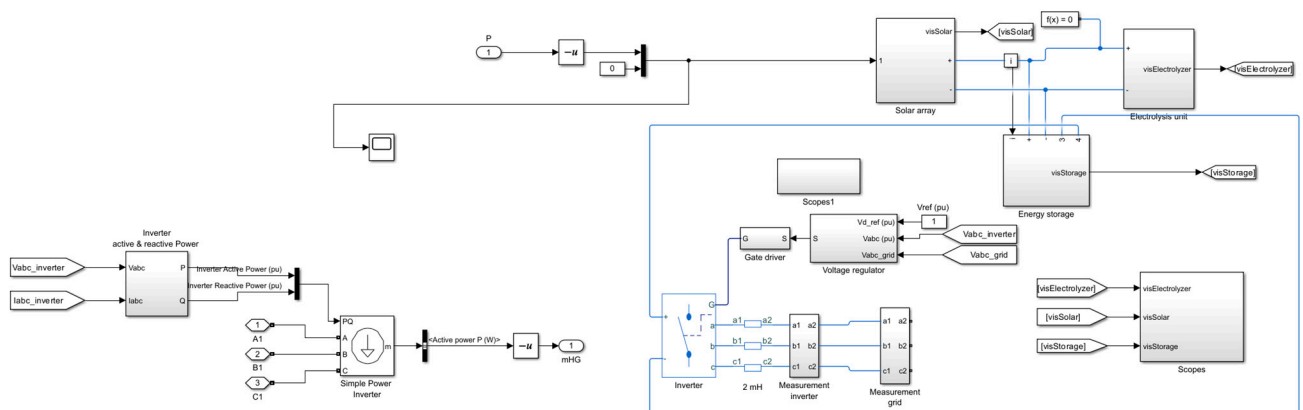

**Figure 3.** Microgrid with EMS Optimization for Solar-Hydrogen.

**Figure 4.** Hybrid Solar-Hydrogen.

### 4.2. Performance Result

The system's effectiveness and efficiency are demonstrated by the performance data based on the created Microgrid ESS Optimization for the Solar-Hydrogen system. In determining the success of the design, the evaluation takes into account a number of essential factors. The overall energy efficiency of the microgrid is the focus of the performance outcomes (Figures 5 and 6). This includes determining how well the system optimizes solar and hydrogen-derived energy storage, delivery, and consumption, with the goal

of minimizing energy waste. The microgrid's performance is measured in terms of reliability and resilience. This includes measuring how well the optimized energy storage system responds to changes in energy demand, changes in solar availability, and external interruptions, ensuring a continuous and stable power supply.

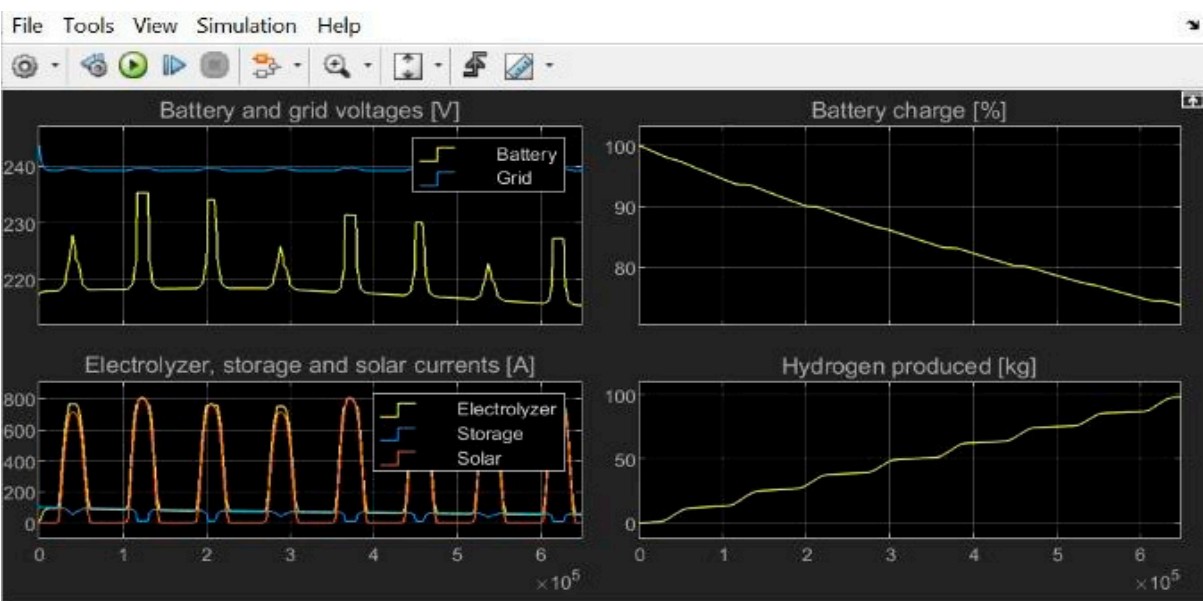

**Figure 5.** Performance Result 1.

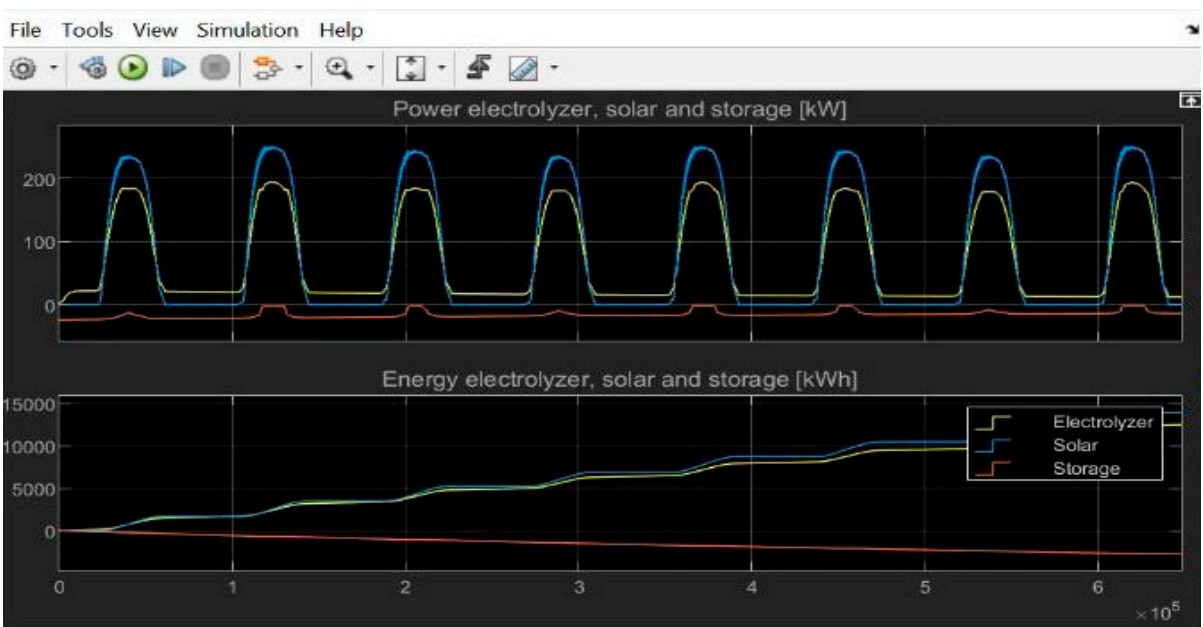

**Figure 6.** Performance Result 2.

The battery and grid voltage Simulink chart findings provide important information about the system's electrical performance (Figure 5). The average grid voltage stabilizes between 239 and 240 volts, indicating a steady and dependable grid power supply. Conversely, the Battery Voltage displays a dynamic range of 218 to 235 volts, indicating the battery's charge-discharge cycles. A discharge phase is indicated by the lowest recorded voltage of 218 and a charging phase or high battery capacity is indicated by the maximum voltage of 235. The system's adaptive nature is reflected in the battery's variability, which enables it to manage demand fluctuations and intermittent renewable energy inputs while

efficiently using the grid as a reliable power source. The Simulink diagram highlights how the grid and battery components work well together to provide a dependable and flexible energy source that stays within the designated voltage range.

The findings of the Simulink chart that shows the solar currents, storage, and electrolyzer give a thorough picture of the system's energy dynamics (Figure 5). The Electrolyzer and Solar Currents are noteworthy for their synchronized patterns, which frequently peak together. A maximum recorded score of 800 amperes was achieved. This coherence implies that the solar panels maximize the production of hydrogen by contributing significantly to the electrolysis process under ideal solar conditions. On the other hand, the storage system's charging and discharging actions are indicated by the Storage Current, which has a maximum score of 100 amperes. The comparatively smaller magnitude when contrasted with the Electrolyzer and Solar Currents suggests that the energy storage and retrieval system is functioning effectively, guaranteeing a consistent and balanced supply of electricity. All things considered, these Simulink chart results highlight how well electrolysis, solar power generation, and energy storage are coordinated, presenting a system that makes the best use of renewable energy sources while preserving a steady and flexible energy supply.

The Battery Charge Percentage Simulink chart result offers important insights into the energy storage system's dynamic behavior (Figure 5). When the battery reaches its maximum performance result of 100%, it indicates that it is fully charged, that available energy sources are being used efficiently, or that there may be a period of low energy demand. On the other hand, the drop to the lowest recorded figure of 30% indicates a situation in which the battery has experienced significant discharge, either during a time of increased energy consumption or restricted production of renewable energy. This variation in the battery charge percentage highlights how adaptable the system is, skillfully controlling the amount of energy stored to satisfy changing needs. The battery's robust functionality is demonstrated by the Simulink chart, which also shows how well it can store and release energy while adapting to the dynamic nature of energy supply and consumption within the given parameters.

The system's impressive ability in producing hydrogen by electrolysis is shown in the Simulink chart result for Hydrogen Production in kilos, where the highest score was 98 (Figure 5). This peak value suggests that the energy resources were used efficiently, most likely when solar power input was at its best and the grid was stable. The system's ability to capture renewable energy sources and transform them into a clean, sustainable energy source is demonstrated by the significant hydrogen generation. As the Simulink chart illustrates, reaching a high level of hydrogen production means that solar power and electrolysis processes have been successfully integrated, which is in line with the system's goals of encouraging green hydrogen generation. This result highlights the system's potential as a dependable and environmentally beneficial part of the larger energy landscape and supports its ability to considerably contribute to the creation of a clean energy source.

The power results for the Electrolyzer, Solar, and Storage systems in kilowatts on the Simulink chart provide important information on the energy dynamics of the system (Figure 6). The Electrolyzer's range is 170 to 190 kW, and the highest score denotes maximal electrolysis activity, perhaps at times when solar power output is abundant. Interestingly, the Solar Power exhibits a steady and coordinated growth with the Electrolyzer, ranging from 220 to 260 kW. According to this alignment, the electrolyzer reacts proportionately to an increase in solar power input, optimizing the production of hydrogen. Nonetheless, the Storage electricity continuously shows less than 0 kW, suggesting that the storage system is more often discharging than producing electricity. This scenario demonstrates the system's skillful control of power flow and storage discharge by implying a planned use of stored energy, possibly during times of decreased solar input or increasing demand. All things considered, the Simulink diagram highlights how the Electrolyzer, Solar, and Storage

components are interdependent and work together to create a dynamic and effective energy balance in the system.

The energy performance of the system is shown dynamically by the Simulink chart results for Energy in kilowatt-hours (kWh) for the Electrolyzer, Solar, and Storage (Figure 6). From 0 to 14,000 kWh, the electrolyzer's energy output increases gradually, suggesting a steady and significant production of hydrogen over time. Notably, the Solar Energy exhibits a coordinated performance pattern with the Electrolyzer, ranging from 0 to 12,000 kWh. The concurrent rise in Electrolyzer and Solar performance scores points to a balanced reaction to ideal solar power conditions, which increases the production of hydrogen. The Energy stored in the Storage component, on the other hand, indicates that stored energy is primarily being used rather than accumulating, as it displays a downward trend from 0 kWh. This use is consistent with efficient energy management; it might even direct stored energy to adjust to changing grid conditions or demand variations. Thus, the Simulink diagram illustrates a flexible and integrated system in which the Solar and Electrolyzer components work together to promote the generation of hydrogen, while the Storage element dynamically controls energy storage and release in response to the system's actual needs.

## 5. Future Prospects

### 5.1. The Development of AIoT (Artificial Intelligence of Things)

The Internet of Things (IoT) for Hydrogen Systems is a layered architecture that integrates AI and the Internet of Things (IoT) to improve the efficiency and efficacy of hydrogen energy systems. Each layer helps to optimize hydrogen production, distribution, and utilization. Table 5 describes in detail the four basic layers of this design.

**Table 5.** Architecture of AIoT for Hydrogen Energy Systems.

| No | Layer | Description & Features |
|---|---|---|
| 1 | Artificial Intelligence | The AI layer is the AIoT system's brain, responsible for data processing, decision-making, and improving hydrogen-related processes. Advanced machine learning, deep learning, and AI algorithms are included. The AI layer improves the system's intelligence by enabling data-driven decision-making, predictive capabilities, and continual development. Features: a. Data Analysis: AI algorithms evaluate data from IoT sensors and devices to find patterns, anomalies, and trends in hydrogen production, storage, and consumption. b. Decision-Making: AI models employ sensor data to make smart decisions in real-time, such as altering hydrogen production rates, maximizing energy use, or initiating safety routines. c. Predictive Maintenance: AI anticipates when equipment and infrastructure need maintenance, decreasing downtime and ensuring system reliability. d. Energy Optimization: AI optimizes energy use and production by matching hydrogen production to demand to reduce costs and environmental impact. This layer offers a framework for efficiently connecting and managing the AI and IoT components. Cloud platforms, communication protocols, and networking infrastructure are all part of it. The platform and network layer provide for seamless data flow across the AI and IoT levels while also providing data security, scalability, and accessibility. Features: |

**Table 5.** *Cont.*

| No | Layer | Description & Features |
|---|---|---|
| 2 | Platform and Network | a. Data Analysis:<br>AI algorithms evaluate data from IoT sensors and devices to find patterns, anomalies, and trends in hydrogen production, storage, and consumption.<br>b. Decision-Making:<br>AI models employ sensor data to make smart decisions in real time, such as altering hydrogen production rates, maximizing energy use, or initiating safety routines.<br>c. Predictive Maintenance:<br>AI anticipates when equipment and infrastructure need maintenance, decreasing downtime and ensuring system reliability.<br>Energy Optimization: AI optimizes energy use and production by matching hydrogen production to demand to reduce costs and environmental impact.<br>d. Data Integration:<br>Collects and transmits data from IoT sensors and devices to the AI layer for analysis and decision-making.<br>Cloud Services: Cloud computing resources are used for data storage, processing, and scalability.<br>e. Secure Communication:<br>Ensures that data is transmitted securely and reliably between IoT devices and the AI layer.<br>Data Management: Data Management is in charge of storing, retrieving, and archiving data for historical analysis and reporting. |
| 3 | IoT Sensors and Devices | This layer is made up of a network of Internet of Things sensors, actuators, and Devices that are deployed across the hydrogen energy system. These gadgets capture data in real-time and interact with physical components.<br>The IoT layer delivers real-world data that flows into the AI layer, allowing for easier monitoring, control, and data-driven decision-making.<br>Features:<br>a. Data Collection:<br>IoT sensors collect data on hydrogen pressure, temperature, and flow rates, energy usage, and other pertinent characteristics.<br>b. Control:<br>In reaction to AI-driven judgments, actuators and devices can remotely control equipment, valves, or processes.<br>c. Environmental Sensors:<br>Monitor environmental factors such as emissions or air quality to ensure regulatory compliance. |
| 4 | Energy Resources and Production | This layer represents the physical infrastructure involved in hydrogen creation, storage, and distribution. Electrolyzers, hydrogen storage tanks, pipes, and other components are included.<br>Features:<br>Energy Resources and Production:<br>This layer represents the physical infrastructure involved in hydrogen creation, storage, and distribution. Electrolyzers, hydrogen storage tanks, pipes, and other components are included.<br>The energy resource and production layer is the hydrogen system's backbone, and IoT sensors in this layer provide essential data for AI-driven optimization and control. |

By integrating these four layers, AIoT for Hydrogen Systems increases the efficiency, safety, and sustainability of hydrogen energy operations, making them more adaptive to dynamic conditions and contributing to the spread of clean hydrogen technologies (Figure 7).

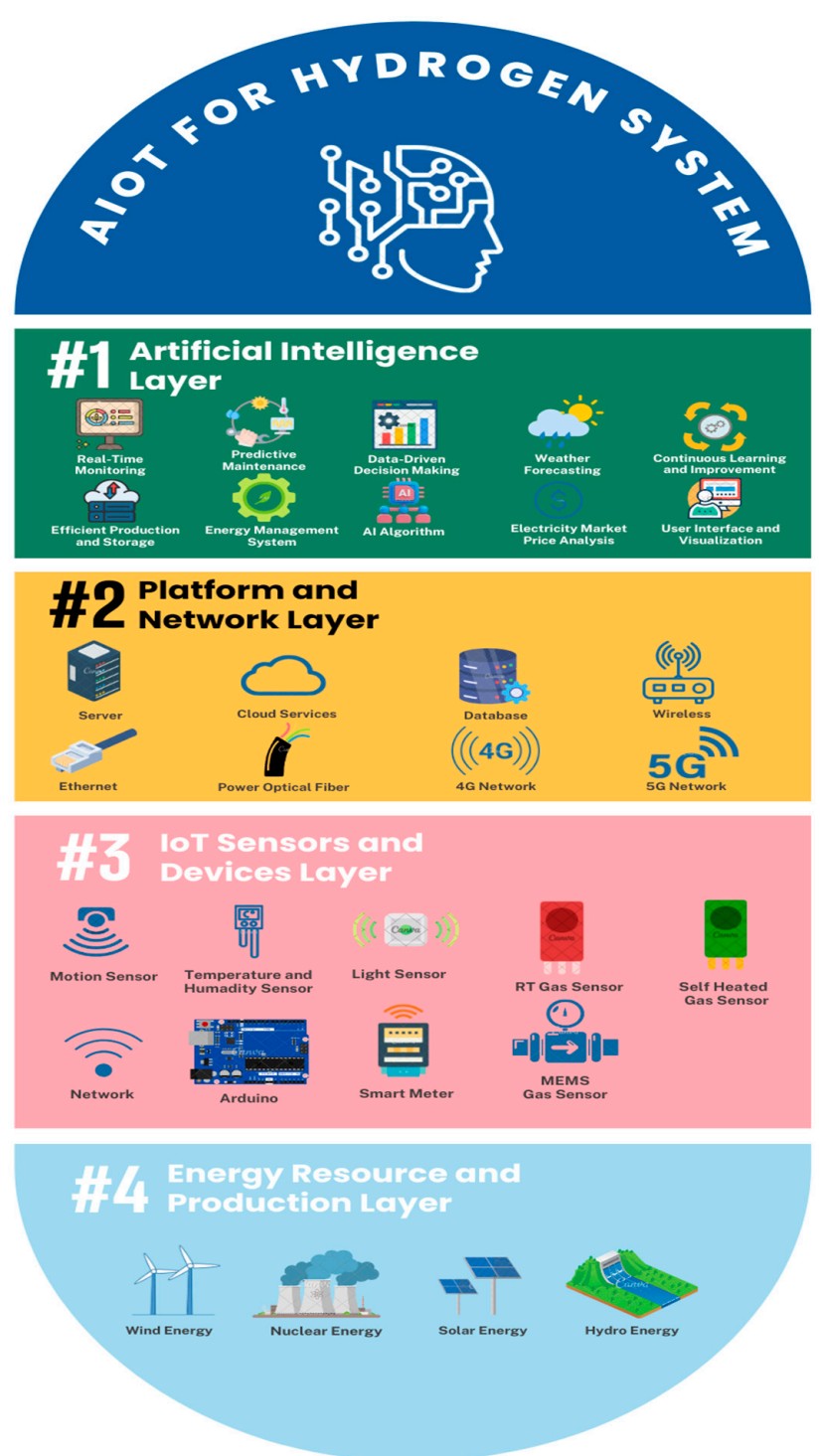

**Figure 7.** The Architecture of AIoT for Hydrogen Energy Systems.

*5.2. AIoT for Hydrogen Energy Systems*

Incorporating artificial intelligence (AI) into hydrogen energy systems (AIoT) is a crucial step toward generating cleaner, more effective, and safer energy solutions. By combining the power of AI algorithms with the interconnection of IoT devices, AIoT technology advances every stage of the hydrogen energy lifecycle, from production and storage to distribution and usage (Figure 8).

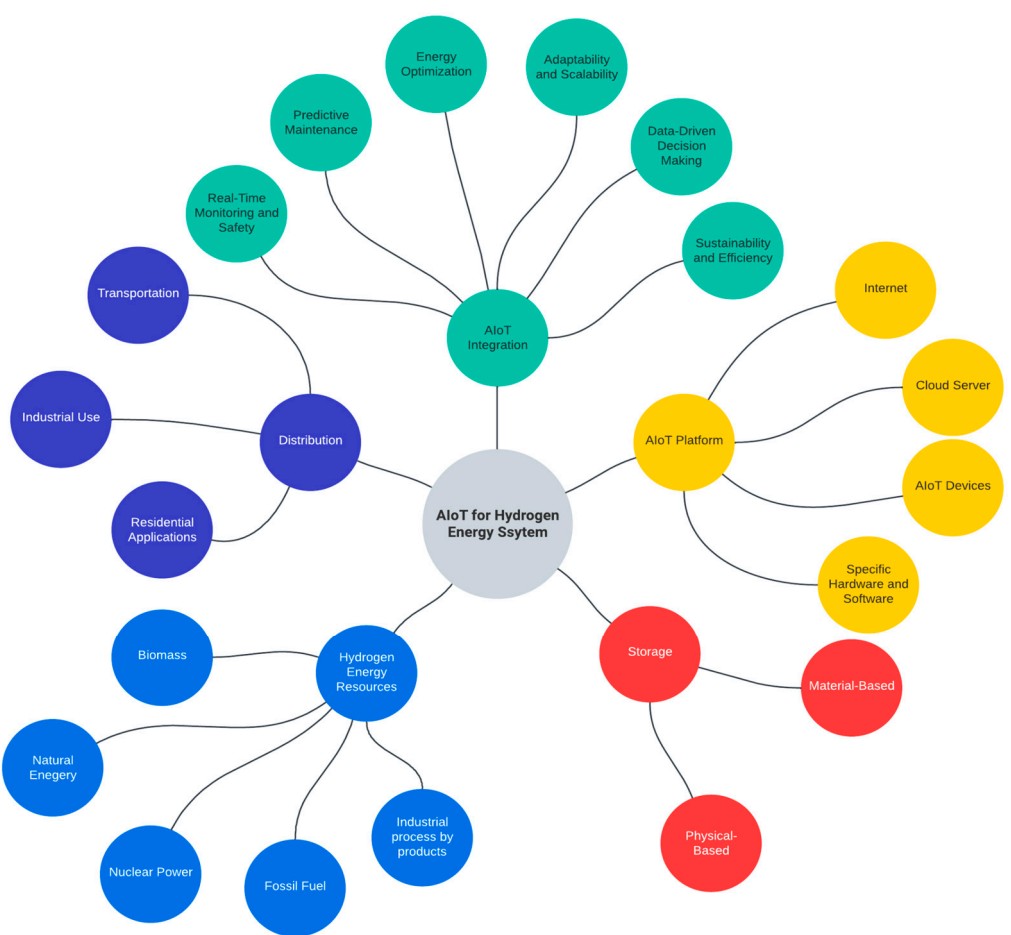

**Figure 8.** Mind Map of AIoT for Hydrogen Energy System.

Because of the AIoT's real-time monitoring and prediction capabilities, proactive maintenance is possible, minimizing risks and downtime while providing rapid responses to safety hazards. Real-time data from renewable sources is used to optimize hydrogen manufacturing operations, resulting in more efficient energy consumption and reduced environmental impact. Furthermore, the AIoT systems' adaptability to changing conditions assures that hydrogen energy systems may evolve in parallel with improvements in technology and energy demand.

## 6. Conclusions

This research explores the unknown field of integrating a solar-hydrogen storage system into the energy management framework of a university. We investigated the theoretical foundations, empirical case studies, and practical implementations of this unique energy solution using a thorough research strategy. The literature assessment illuminated the current knowledge landscape, highlighting gaps and laying the groundwork for our investigation. Real-world case studies and expert interviews provided essential insights into the challenges, achievements, and lessons learned from comparable undertakings, rooting our research in the reality of implementation.

The technological evaluation examined solar-hydrogen system components, evaluating their efficiency, scalability, and economic viability. We projected the system's performance under various scenarios using simulation tools and modeling methodologies, providing a predictive peek into its behavior. Our environmental impact analysis examined the ecological footprint, life cycle evaluations, and carbon emission reductions to ensure that the proposed integration complies with the university's greater goal of sustainability.

These varied ideas were distilled into a workable plan through integration design and feasibility research. We laid out a step-by-step plan for seamlessly integrating the solar-hydrogen storage system into the university's existing energy infrastructure. This design takes into account not only technical complexities but also economic aspects and prospective problems, providing a comprehensive view of the viability and practicality of our suggested solution. Our research contributes to the expanding landscape of sustainable energy solutions in educational institutions and beyond as we envision the adoption of this solar-hydrogen integration. Finally, this investigation goes beyond theoretical principles by providing a concrete pathway to a greener and more resilient energy future for universities and other organizations seeking to embrace cutting-edge technologies and environmental stewardship.

**Author Contributions:** S.R.J.: project evaluation, methodology, investigation, resources, supervision, modeling, simulation S.P.: software development, functionality evaluation. K.K.: conceptualization, funding acquisition, resources, supervision, writing—original draft, writing—review and editing. All authors have read and agreed to the published version of the manuscript.

**Funding:** This research was supported by "Regional Innovation Strategy (RIS)" through the National Research Foundation of Korea (NRF), funded by the Ministry of Education (MOE) (2022RIS-005).

**Institutional Review Board Statement:** Not applicable.

**Informed Consent Statement:** Not applicable.

**Data Availability Statement:** The original contributions presented in the study are included in the article, further inquiries can be directed to the corresponding authors.

**Conflicts of Interest:** The authors declare no conflicts of interest.

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
