# Peer review of "H2 URESONIC: Design of a Solar-Hydrogen University Renewable Energy System for a New and Innovative Campus"

_applsci, doi:10.3390/app14041554_

Round 1

Reviewer 1 Report

Comments and Suggestions for Authors

This article investigates the feasibility of integrating a solar-hydrogen storage system into university energy management. The study contextualizes the growing importance of renewable energy in universities, emphasizing the need for efficient energy storage. It explores solar-hydrogen technology, detailing its components, operation, and advantages over conventional storage systems. The Integration Design chapter assesses existing university energy infrastructures, addresses challenges, and proposes seamless integration strategies, considering technological and economic factors. Real-world case studies with performance metrics offer valuable insights. The Future Prospects section discusses emerging trends and government impact, providing a forward-looking perspective. After reading the manuscript several times, the reviewer found the following points need to be revised to make the paper qualified to be published in MDPI Applied Sciences:

1.     The benefits of solar-hydrogen integration are emphasized. However, to augment the persuasiveness of the argument, it is recommended to incorporate more quantified data analyses in the form of figures.

2.     The introductory section lacks adequate context regarding the significance and challenges of integrating a solar-hydrogen storage system into university energy management. Further exploration of existing literature on this subject is recommended to enhance the foundation of the article. Are there any published papers that specifically address this topic?

3.     Could the variables presented in lines 130 to 169 be more effectively organized into distinct categories for improved coherence and conciseness, rather than being presented in a linear list

4.     Figures 2, 3, and 8 suffer from readability issues attributed to a small font size.

5.     The term "Microgrid ESS" in line 386 lacks a clear definition based on the existing body of published literature.

6.     The manuscript references Figure 4, yet its explanation is not sufficiently elucidated.

7.     The captions for Figures 5 and 6 are overly simplistic and may benefit from more detailed and informative descriptions.

8.     Table 4 in line 423 appears to lack effective organization.

Author Response

For research article

Response to Reviewer 1 Comments

This article investigates the feasibility of integrating a solar-hydrogen storage system into university energy management. The study contextualizes the growing importance of renewable energy in universities, emphasizing the need for efficient energy storage. It explores solar-hydrogen technology, detailing its components, operation, and advantages over conventional storage systems. The Integration Design chapter assesses existing university energy infrastructures, addresses challenges, and proposes seamless integration strategies, considering technological and economic factors. Real-world case studies with performance metrics offer valuable insights. The Future Prospects section discusses emerging trends and government impact, providing a forward-looking perspective. After reading the manuscript several times, the reviewer found the following points need to be revised to make the paper qualified to be published in MDPI Applied Sciences:

1. The Benefits

The benefits of solar-hydrogen integration are emphasized. However, to augment the persuasiveness of the argument, it is recommended to incorporate more quantified data analyses in the form of figures. [This research is preliminary research which is an initial conceptualization for the needs of research progress. In the meantime, we can present data in the form of an analysis of the existence of the system and the proposal and modeling in the form of a simulation (Simulink) to show that the proposed system is being implemented..]

2. The Introduction

The introductory section lacks adequate context regarding the significance and challenges of integrating a solar-hydrogen storage system into university energy management. Further exploration of existing literature on this subject is recommended to enhance the foundation of the article. Are there any published papers that specifically address this topic?. [We haven't found any discussions at the University, but there are several related articles discussing energy management issues which we use as references in our articles

1.             Bernhard, T.; Stefan, P.; Andreas, W.; Gerhard, P. Hybrid model predictive control of renewable microgrids and seasonal hydrogen storage. International Journal of Hydrogen Energy. 2023, 48, 38125-38142.

2.             Albert, H. S.; Arjen, A.T. V.; Michiel, A. J. U. H. B.; Evrim, U. A Green Hydrogen Energy System: Optimal control strategies for integrated hydrogen storage and power generation with wind energy. Renewable and Sustainable Energy Reviews. 2022, 168, 112744, 1-14.

3.             Vincent, K.; Veronika, E.; Kerstin, A.; Stefan, R.; Andrea, H. Decentral Production of Green Hydrogen for Energy Systems: An Economically and Environmentally Viable Solution for Surplus Self-Generated Energy in Manufacturing Companies? Sustainability MDPI.  2023, 15, 4, 1-27.

4.             Ahmed, M. A. O.; Mohamed, G. O.; Bishoy, E. S. Evaluation of green hydrogen production using solar, wind, and hybrid technologies under various technical and financial scenarios for multi-sites in Egypt. International Journal of Hydrogen Energy. 2023, 48, 38535-38556.

5.             Ismail, M.; Tawfik, G.; Badr, M. A.; Khalid, A.; Ahmed, A.; Mansor, A.; Hsan, H. A. Integration of Renewable-Energy-Based Green Hydrogen into the Energy Future. Processes MDPI. 2023, 11, 9, 1-29.

6.             Babangida, M.; Md P. A.; Abba, L. B.; Mukhtar, F. H. A systematic review of hybrid renewable energy systems with hydrogen storage: Sizing, optimization, and energy management strategy. International Journal of Hydrogen Energy. 2023, 48, 97, 38354-38373.

7.             Torbjørn, E. E.; Amin H.; Sabrina, S. Hydrogen-based systems for integration of renewable energy in power systems: Achievements and perspectives. International Journal of Hydrogen Energy. 2021, 46, 31963-31983.

8.             Wilson, F. E.; Joseph, T. A.; Martins, C. O.; Philemon, C. U. A Solar Energy System Design for Green Hydrogen Production in South-Western Nigeria, Lagos State, Using HOMER & ASPEN. Open Journal of Optimization. 2023, 12, 72-97.

9.             Zheng, L.; Wenda, Z.; Rui, Z.; Hexu, S. Development of renewable energy multi-energy complementary hydrogen energy system (A Case Study in China): A review. Energy Exploration & Exploitation. 2020, 38, 6, 2099-2127

3. Lines 130 to 169

Could the variables presented in lines 130 to 169 be more effectively organized into distinct categories for improved coherence and conciseness, rather than being presented in a linear list. [Updated and Organized in Table 4]

4. Figures 2, 3, and 8

Figures 2, 3, and 8 suffer from readability issues attributed to a small font size.. [(Updated, Figure 2, 3, 4 and 8)]

5. The term "Microgrid ESS"

The term "Microgrid ESS" in line 386 lacks a clear definition based on the existing body of published literature. [Revised as EMS (Energy Management System)]

6. Explanation Figure 4

The manuscript references Figure 4, yet its explanation is not sufficiently elucidated. [Updated and Added in Lines 361-371].

7. Figures 5 and 6

The captions for Figures 5 and 6 are overly simplistic and may benefit from more detailed and informative descriptions. [The explanation is on lines 379-388. The limited explanation is because we have not carried out further analysis of the expected results which we will carry out at the next stage of the research.].

8. Table 4

Table 4 in line 423 appears to lack effective organization. [(Organized and Updated as a Table 5)].

Reviewer 2 Report

Comments and Suggestions for Authors

This paper reports the authors’ proposal of a solar-hydrogen storage system for the energy management of a university. The authors declare that the major goal of their research is to offer a thorough examination of the solar-hydrogen storage system to support universities in making informed decisions about adopting and integrating this technology into their energy infrastructure. However, the submitted manuscript only outlines the solar-hydrogen storage system, components, monitoring system, management system model and a very brief description of the simulation result and future prospects. Also, the abstract and Introduction mention about;1. benefits over typical storage systems,
2. identification of problems for system integration that will be solved by a cost-benefit analysis and scalability studies,
3. case studies include real-world examples.
However, I could not find enough information about them.
For instance, benefit over typical systems should be discussed in Introduction, perhaps in Table 1 with a clearer comparison between other technologies, such as simple PV-battery storage systems, with a viewpoint for a system integration to the university. Otherwise, readers can not understand why a solar-hydrogen storage system is suitable for university integration purposes.
As for the 2nd point, cost-benefit analysis and scalability studies would be addressed in chapter 4. But I could not find clear descriptions on these studies.
The 3rd point would be addressed in chapter 3 for modeling the system for Kangwon National University Samcheok Campus. But there is no basic information for designing the system, such as current energy demand/supply information. So I can not understand their targeting performance of the proposed system and how they optimize the system.
In addition, there is no explanation for “H2 URESONIC”.
In conclusion, I recommend the paper to be rejected."

Author Response

For research article

Response to Reviewer 2 Comments

This paper reports the authors’ proposal of a solar-hydrogen storage system for the energy management of a university. The authors declare that the major goal of their research is to offer a thorough examination of the solar-hydrogen storage system to support universities in making informed decisions about adopting and integrating this technology into their energy infrastructure. However, the submitted manuscript only outlines the solar-hydrogen storage system, components, monitoring system, management system model and a very brief description of the simulation result and future prospects. Also, the abstract and Introduction mention about;1. benefits over typical storage systems, identification of problems for system integration that will be solved by a cost-benefit analysis and scalability studies,

1. Case Study

Case studies include real-world examples. However, I could not find enough information about them. For instance, benefit over typical systems should be discussed in Introduction, perhaps in Table 1 with a clearer comparison between other technologies, such as simple PV-battery storage systems, with a viewpoint for a system integration to the university. Otherwise, readers can not understand why a solar-hydrogen storage system is suitable for university integration purposes. As for the 2nd point, cost-benefit analysis and scalability studies would be addressed in chapter. [The research we carried out was preliminary research that received funding from the Regional Innovation Strategy (RIS) through the National Research Foundation of Korea (NRF), funded by the Ministry of Education (MOE) (2022RIS-005).

Where in our project as a whole we collaborate with the government of the city of Samcheok, Gangwon-Do which is also an example city for Hydrogen energy. So in the discussion of writing this research, what is emphasized is the idea of implementing Hydrogen energy at Kangwon National University, which is a university located in Samcheok City as well.

We have updated several parts based on suggestions from other reviewers to make it easier for readers to understand our writing. We thank you for your suggestions and comments..]

2. Chapter 3

But I could not find clear descriptions on these studies.

The 3rd point would be addressed in chapter 3 for modeling the system for Kangwon National University Samcheok Campus. But there is no basic information for designing the system, such as current energy demand/supply information. So I can not understand their targeting performance of the proposed system and how they optimize the system.. [In section 3.2 (Lines 306-348) Integration Design we have explained the energy sources and location of solar energy sources which we used as a case study. In this article we did not carry out the energy needs analysis stage because the research scope emphasizes analyzing the situation in the form of available energy sources and designs for implementing a Solar-Hydrogen system that could be implemented on the Samcheok Campus.]

3. Explanation for “H2 URESONIC”.

In addition, there is no explanation for “H2 URESONIC”.. [H2 URESONIC is the abbreviation that is stated in the title of our research, where the meaning of H2 URESONIC is Hydrogen University Renewable Energy System for New and Innovative Campus. We chose this name because of the vision of using hydrogen energy on the Kangwon National University campus which is located in Samcheok City, where Samcheok City has the name H2 DREAM.]

Round 2

Reviewer 1 Report

Comments and Suggestions for Authors

The study commences by contextualizing the growing relevance of renewable energy in universities and underscores the critical need for efficient energy storage systems. Its primary focus is the in-depth exploration of solar-hydrogen technology, covering its components, operational mechanisms, and advantages compared to traditional storage systems. The Integration Design chapter meticulously examines current university energy infrastructures, identifies challenges, and proposes solutions for seamlessly integrating solar-hydrogen systems, emphasizing technological and economic considerations through cost-benefit analyses and scalability studies. Real-world case studies provide valuable insights from successful implementations. The Future Prospects chapter explores emerging trends in solar-hydrogen technology and considers the impact of government legislation, offering a forward-looking viewpoint for colleges. The report concludes with a concise summary of significant findings, highlighting the benefits of solar-hydrogen integration and providing recommendations for future implementations. After careful consideration, the reviewer deems the paper worthy of publication in MDPI Applied Sciences.

Author Response

For research article

Response to Reviewer 1 Comments

The study commences by contextualizing the growing relevance of renewable energy in universities and underscores the critical need for efficient energy storage systems. Its primary focus is the in-depth exploration of solar-hydrogen technology, covering its components, operational mechanisms, and advantages compared to traditional storage systems.

1. The Research

The Integration Design chapter meticulously examines current university energy infrastructures, identifies challenges, and proposes solutions for seamlessly integrating solar-hydrogen systems, emphasizing technological and economic considerations through cost-benefit analyses and scalability studies. Real-world case studies provide valuable insights from successful implementations. The Future Prospects chapter explores emerging trends in solar-hydrogen technology and considers the impact of government legislation, offering a forward-looking viewpoint for colleges. The report concludes with a concise summary of significant findings, highlighting the benefits of solar-hydrogen integration and providing recommendations for future implementations. After careful consideration, the reviewer deems the paper worthy of publication in MDPI Applied Sciences.:. [We are truly appreciated for your comments and suggestions, we believe that our work still have limitations. We are looking to develop and make a good work in the future and especially can give a beneficial contribution to the community in terms of solar-hydrogen energy]

Reviewer 2 Report

Comments and Suggestions for Authors

The revised version (and the authors’ replies) did not fully answer my comments. Their study is under "the idea of implementing Hydrogen energy at Kangwon National University located in Samcheok City" which is promoting Hydrogen energy. Therefore, there is no other comparison or benefits study that relates to my comment 1. The authors’ situation (restrictions) can be understandable; however, what new or significant information can this manuscript give to the readers? Of course, to contribute to the discussion of sustainable energy in educational institutions with this study, which focuses on the architecture and integration design of solar hydrogen storage systems. However, the revised version of the paper does not include enough information to discuss even in educational institutions.

My 2nd comment is where is the discussion for identifying problems for this system integration. However, the revised version still only has their simulation results without any discussion part. Also, with Figures 5 and 6, who can understand “how well the system optimizes solar and hydrogen-derived energy storage, delivery, and consumption, with the goal of minimizing energy waste”? There must be explanations of what these figures express, at least.

The authors answered my 3rd comment “In this article we did not carry out the energy needs analysis stage because the research scope emphasizes analyzing the situation in the form of available energy sources and designs for implementing a Solar-Hydrogen system that could be implemented on the Samcheok Campus.” As I commented above, I see the authors’ situation, and the research scope is limited to how the hydrogen system can be integrated into the existing university’s energy system. If so, the abstract and introduction sections have to clearly describe their research scope.

In conclusion, I still can not accept the revised version of the paper.

Author Response

For research article

Response to Reviewer 2 Comments

The revised version (and the authors’ replies) did not fully answer my comments. Their study is under "the idea of implementing Hydrogen energy at Kangwon National University located in Samcheok City" which is promoting Hydrogen energy. Therefore, there is no other comparison or benefits study that relates to my comment,

1. Discussion related to research

1. The authors’ situation (restrictions) can be understandable; however, what new or significant information can this manuscript give to the readers? Of course, to contribute to the discussion of sustainable energy in educational institutions with this study, which focuses on the architecture and integration design of solar hydrogen storage systems. However, the revised version of the paper does not include enough information to discuss even in educational institutions.

[Updated in lines 53-95.

In this research we are considered about three aspects to support our research:

1. Environmental Factors

The development of solar-hydrogen energy presents a viable and sustainable approach to address the negative environmental effects associated with traditional energy sources. This method encourages the production of carbon-neutral energy by using solar electricity to electrolyze hydrogen. When operating, solar-hydrogen systems emit no greenhouse gases, in contrast to fossil fuels, which helps to reduce air pollution and mitigate climate change. Additionally, by providing a flexible and environmentally benign substitute in industries like transportation, manufacturing, and power generation, hydrogen as a clean energy carrier aids in the shift to a low-carbon economy. Adopting solar-hydrogen energy supports a cleaner and more sustainable energy future and is in line with international efforts to stop environmental damage.

(Cited from: Troy Stangarone, 2021, “South Korean eforts to transition to a hydrogen economy”, Clean Technologies and Environmental Policy (2021) 23:509–516 https://doi.org/10.1007/s10098-020-01936-6)

2. Clean and Sustainable Energy

Solar-hydrogen energy combines solar power with electrolysis to produce hydrogen, thereby emulating a clean and sustainable energy paradigm. Sunlight is captured by solar panels and transformed into electricity, which is then used to split water molecules and produce hydrogen, an environmentally friendly fuel. This technology offers a carbon-neutral substitute for traditional energy sources because it produces no emissions at all. The intermittent nature of solar electricity can be mitigated with hydrogen, an energy carrier that is both flexible and storable. Furthermore, hydrogen is only produced by water vapor when it is burned or used in fuel cells, confirming its status as a clean and sustainable energy source. This novel combination of solar and hydrogen technologies not only lessens its negative effects on the environment but also establishes solar-hydrogen energy as a key participant in the shift to a more sustainable and environmentally friendly energy landscape.

(Cited from: Layth Hazim Majida, Hamid Hazim Majidb, Hussein Fawzi Husseinc, 2018, “Analysis of Renewable Energy Sources, Aspects of Sustainability and Attempts of Climate Change”, American Scientific Research Journal for Engineering, Technology, and Sciences (ASRJETS) ISSN (Print) 2313-4410, ISSN (Online) 2313-4402)

3. Example Integration of electrolytic systems of hydrogen generation in Silesian University of Technology

Numerous advantages arise from using electrolytic systems in universities to produce hydrogen, supporting both sustainability and academic goals. Universities can achieve environmental stewardship and lower their carbon footprint by producing clean hydrogen on-site through the use of electrolyzers driven by renewable energy sources like solar or wind. Furthermore, incorporating electrolytic systems into academic environments provides a useful and instructive example of cutting-edge sustainable technology. In order to promote an innovative and environmentally conscious culture, staff and students get personal experience with hydrogen production methods and applications of renewable energy. Moreover, the hydrogen generated may be used as a flexible energy source for a range of campus uses, such as hydrogen fuel for automobiles, offering a real-world illustration of how sustainable energy solutions can be incorporated into regular operations.

(Cited from: Tadeusz Chmielniak, 2019 “Wind and solar energy technologies of hydrogen production – a review of issues” Polityka Energetyczna – Energy Policy Journal 2019  Volume 22  Issue 4  5–20 DOI: 10.33223/epj/114755)

The current study examines the scientific and financial feasibility of powering Delhi Technological University's Science Block in Delhi, India, utilizing solar photovoltaic energy sources. In a hybrid energy system, the intermittent solar energy is stabilized by using hydrogen energy storage to produce a steady electrical current. The most ideal design, with a net current cost of $1,030,406, has been simulated by HOMER software. With a rated capacity of 240 kW and a mean output of 44.5 kW—or 1,068 kWh/day—solar photovoltaic generates 389.865 MWh of energy every year. When operating for 4,344 hours a year, the PV system can produce up to 253 kW, with a PV penetration of 358%.

(Cited from: Alfred John, Srijit Basu, Akshay, Anil Kumar, 2021 “Design and evaluation of stand-alone solar-hydrogen energy storage system for academic institute: A case study”, Materialstoday: Proceedings, Volume 47, Part 17, 2021, Pages 5918-5922)

2. Figure 5 and 6

My 2nd comment is where is the discussion for identifying problems for this system integration. However, the revised version still only has their simulation results without any discussion part. Also, with Figures 5 and 6, who can understand “how well the system optimizes solar and hydrogen-derived energy storage, delivery, and consumption, with the goal of minimizing energy waste”? There must be explanations of what these figures express, at least.. [Updated in lines 432-512

Figure 5

Battery and Grid Voltages [V]

The battery and grid voltage Simulink chart findings provide important information about the system's electrical performance. The average grid voltage stabilizes between 239 and 240 volts, indicating a steady and dependable grid power supply. Conversely, the Battery Voltage displays a dynamic range of 218 to 235 volts, indicating the battery's charge-discharge cycles. A discharge phase is indicated by the lowest recorded voltage of 218 and a charging phase or high battery capacity is indicated by the maximum voltage of 235. The system's adaptive nature is reflected in the battery's variability, which enables it to manage demand fluctuations and intermittent renewable energy inputs while efficiently using the grid as a reliable power source. The Simulink diagram highlights how the grid and battery components work together well to provide a dependable and flexible energy source that stays within the designated voltage range.

Electrolyzer, storage and solar currents [A]

The findings of the Simulink chart that shows the solar currents, storage, and electrolyzer give a thorough picture of the system's energy dynamics. The Electrolyzer and Solar Currents are noteworthy for their synchronized pattern, which frequently peaks together. The maximum recorded score of 800 amperes was achieved. This coherence implies that the solar panels maximize the production of hydrogen by contributing significantly to the electrolysis process under ideal solar conditions. On the other hand, the storage system's charging and discharging actions are indicated by the Storage Current, which has a maximum score of 100 amperes. The comparatively smaller magnitude when contrasted with the Electrolyzer and Solar Currents suggests that the energy storage and retrieval system is functioning effectively, guaranteeing a consistent and balanced supply of electricity. All things considered, these Simulink chart results highlight how well electrolysis, solar power generation, and energy storage are coordinated, presenting a system that makes the best use of renewable energy sources while preserving a steady and flexible energy supply.

Battery charge [%]

The Battery Charge Percentage Simulink chart result offers important insights into the energy storage system's dynamic behavior. When the battery reaches its maximum performance result of 100%, it indicates that it is fully charged, that available energy sources are being used efficiently, or that there may be a period of low energy demand. On the other hand, the drop to the lowest recorded figure of 30% indicates a situation in which the battery has experienced significant discharge, either during a time of increased energy consumption or restricted production of renewable energy. This variation in the battery charge percentage highlights how adaptable the system is, skillfully controlling the amount of energy stored to satisfy changing needs. The battery's robust functionality is demonstrated by the Simulink chart, which also shows how well it can store and release energy while adapting to the dynamic nature of energy supply and consumption within the given parameters.

Hydrogen Produces [kg]

The system's impressive ability in producing hydrogen by electrolysis is shown in the Simulink chart result for Hydrogen Production in kilos, where the highest score was 98. This peak value suggests that the energy resources were used efficiently, most likely when solar power input was at its best and the grid was stable. The system's ability to capture renewable energy sources and transform them into a clean, sustainable energy source is demonstrated by the significant hydrogen generation. As the Simulink chart illustrates, reaching a high level of hydrogen production means that solar power and electrolysis processes have been successfully integrated, which is in line with the system's goals of encouraging green hydrogen generation. This result highlights the system's potential as a dependable and environmentally beneficial part of the larger energy landscape and supports its ability to considerably contribute to the creation of a clean energy source.

Figure 6

Power Electrolyzer, solar and storage [kW]

The power results for the Electrolyzer, Solar, and Storage systems in kilowatts on the Simulink chart provide important information on the energy dynamics of the system. The Electrolyzer's range is 170 to 190 kW, and the highest score denotes maximal electrolysis activity, perhaps at times when solar power output is abundant. Interestingly, the Solar Power exhibits a steady and coordinated growth with the Electrolyzer, ranging from 220 to 260 kW. According to this alignment, the electrolyzer reacts proportionately to an increase in solar power input, optimizing the production of hydrogen. Nonetheless, the Storage electricity continuously shows less than 0 kW, suggesting that the storage system is more often discharging than producing electricity. This scenario demonstrates the system's skillful control of power flow and storage discharge by implying a planned use of stored energy, possibly during times of decreased solar input or increasing demand. All things considered, the Simulink diagram highlights how the Electrolyzer, Solar, and Storage components are interdependent and work together to create a dynamic and effective energy balance in the system.

Energy Electrolyzer, solar and storage [kWh]

The energy performance of the system is shown dynamically by the Simulink chart results for Energy in kilowatt-hours (kWh) for the Electrolyzer, Solar, and Storage. From 0 to 14,000 kWh, the electrolyzer's energy output increases gradually, suggesting a steady and significant production of hydrogen over time. Notably, the Solar Energy exhibits a coordinated performance pattern with the Electrolyzer, ranging from 0 to 12,000 kWh. The concurrent rise in Electrolyzer and Solar performance scores points to a balanced reaction to ideal solar power conditions, which increases the production of hydrogen. The Energy stored in the Storage component, on the other hand, indicates that stored energy is primarily being used rather than accumulating, as it displays a downward trend from 0 kWh. This use is consistent with efficient energy management; it might even direct stored energy to adjust to changing grid conditions or demand variations. Thus, the Simulink diagram illustrates a flexible and integrated system in which the Solar and Electrolyzer components work together to promote the generation of hydrogen, while the Storage element dynamically controls energy storage and release in response to the system's actual needs..]

3.Absract and Introduction

3rd comment “In this article we did not carry out the energy needs analysis stage because the research scope emphasizes analyzing the situation in the form of available energy sources and designs for implementing a Solar-Hydrogen system that could be implemented on the Samcheok Campus.” As I commented above, I see the authors’ situation, and the research scope is limited to how the hydrogen system can be integrated into the existing university’s energy system. If so, the abstract and introduction sections have to clearly describe their research scope.”..

[Abstract updated in lines 28-29 “which the limitation of this research is that it only focuses on design and simulation as a phase of preliminary study.”

Introduction updated in lines 53-95

The development of solar-hydrogen energy presents a viable and sustainable approach to address the negative environmental effects associated with traditional energy sources. This method encourages the production of carbon-neutral energy by using solar electricity to electrolyze hydrogen. When operating, solar-hydrogen systems emit no greenhouse gases, in contrast to fossil fuels, which helps to reduce air pollution and mitigate climate change. Additionally, by providing a flexible and environmentally benign substitute in industries like transportation, manufacturing, and power generation, hydrogen as a clean energy carrier aids in the shift to a low-carbon economy. Adopting solar-hydrogen energy supports a cleaner and more sustainable energy future and is in line with international efforts to stop environmental damage [12].

Solar-hydrogen energy combines solar power with electrolysis to produce hydrogen, thereby emulating a clean and sustainable energy paradigm. Sunlight is captured by solar panels and transformed into electricity, which is then used to split water molecules and produce hydrogen, an environmentally friendly fuel. This technology offers a carbon-neutral substitute for traditional energy sources because it produces no emissions at all. The intermittent nature of solar electricity can be mitigated with hydrogen, an energy carrier that is both flexible and storable. Furthermore, hydrogen is only produced by water vapor when it is burned or used in fuel cells, confirming its status as a clean and sustainable energy source. This novel combination of solar and hydrogen technologies not only lessens its negative effects on the environment but also establishes solar-hydrogen energy as a key participant in the shift to a more sustainable and environmentally friendly energy landscape [13]

Numerous advantages arise from using electrolytic systems in universities to produce hydrogen, supporting both sustainability and academic goals. Universities can achieve environmental stewardship and lower their carbon footprint by producing clean hydrogen on-site through the use of electrolyzers driven by renewable energy sources like solar or wind. Furthermore, incorporating electrolytic systems into academic environments provides a useful and instructive example of cutting-edge sustainable technology. In order to promote an innovative and environmentally conscious culture, staff and students get personal experience with hydrogen production methods and applications of renewable energy. Moreover, the hydrogen generated may be used as a flexible energy source for a range of campus uses, such as hydrogen fuel for automobiles, offering a real-world illustration of how sustainable energy solutions can be incorporated into regular operations [14].

The current study examines the scientific and financial feasibility of powering Delhi Technological University's Science Block in Delhi, India, utilizing solar photovoltaic energy sources. In a hybrid energy system, the intermittent solar energy is stabilized by using hydrogen energy storage to produce a steady electrical current. The most ideal design, with a net current cost of $1,030,406, has been simulated by HOMER software. With a rated capacity of 240 kW and a mean output of 44.5 kW—or 1,068 kWh/day—solar photovoltaic generates 389.865 MWh of energy every year. When operating for 4,344 hours a year, the PV system can produce up to 253 kW, with a PV penetration of 358% [15].

Round 3

Reviewer 2 Report

Comments and Suggestions for Authors

The 2nd revision of the manuscript responded to reviewers' comments and suggestions. This paper aims to analyze the available energy sources and designs for implementing a Solar-Hydrogen system on the Samcheok Campus, Kangwon National University located in Samcheok City whose government is promoting Hydrogen energy. The performance of the proposed system integrated with the existing energy system was confirmed by Simulink model simulation. The paper includes basic information of the Solar-Hydrogen system. Therefore, I agree to accept this revised version of the paper.